# Feasibility of Employing mHealth in Delivering Preventive Nutrition Interventions Targeting the First 1000 Days of Life: Experiences from a Community-Based Cluster Randomised Trial in Rural Bangladesh

**DOI:** 10.3390/nu16203429

**Published:** 2024-10-10

**Authors:** Tarana E Ferdous, Md. Jahiduj Jaman, Abu Bakkar Siddique, Nadia Sultana, Takrib Hossain, Shams El Arifeen, Sk Masum Billah

**Affiliations:** 1Maternal and Child Health Division, icddr,b, Dhaka 1212, Bangladesh; mdjahiduj@icddrb.org (M.J.J.); abu.bakkar@icddrb.org (A.B.S.); nadia.s@icddrb.org (N.S.); takrib.hossain@icddrb.org (T.H.); shams@icddrb.org (S.E.A.); sk.billah@sydney.edu.au (S.M.B.); 2Sydney School of Public Health, University of Sydney, Sydney, NSW 2006, Australia

**Keywords:** mHealth, nutrition intervention, community health worker, rural Bangladesh

## Abstract

Background/Objectives: An Android platform-based customised app and web-linked system was developed to aid in implementing selected nutrition interventions by community health workers (CHWs) in a community-based cluster randomised trial (c-RCT) in rural Bangladesh. Methods: Here, we describe the architecture of the intervention delivery system, and explore feasibility of employing mHealth as CHWs’ job aid, employing a mixed-method study design covering 17 visits per mother-child dyad. We analysed CHWs’ real-time visit information from monitoring and documentation data, and CHWs’ qualitative interviews to explore the advantages and barriers of using mHealth as a job aid. Results: Intervention coverage was high across the arms (>90%), except around the narrow perinatal period (51%) due to mothers’ cultural practice of moving to their parents’ homes and/or hospitals for childbirth. CHWs mentioned technical and functional advantages of the job aid including device portability, easy navigability of content, pictorial demonstration that improved communication, easy information entry, and automated daily scheduling of tasks. Technical challenges included charging tablets, especially in power cut-prone areas, deteriorated battery capacity over continuous device usage, unstable internet network in unsupportive weather conditions, and device safety. Nevertheless, onsite supervision and monitoring by expert supervisors remained important to ensure intervention quality. Conclusions: With appropriate training and supervision, CHWs utilised the tablet-based app proficiently, attaining high coverage of long-term visits. mHealth was thus useful for designing, planning, scheduling, and delivering nutrition interventions through CHWs, and for monitoring and supervision by supervisors. Therefore, this application and job aid can be adopted or replicated into the currently developing national health systems platform for improving coverage and quality of preventive maternal and child nutrition services. In addition, continuous supportive supervision by skilled supervisors must be accompanied to ensure CHWs’ task quality. Finally, future studies should rigorously assess undesirable health and environmental effects of mHealth before and after mainstreaming, effective interventions addressing device-induced health hazards should be designed and scaled up, and effective e-waste management must be ensured.

## 1. Introduction

Nutrition is directly addressed in Sustainable Development Goal 2 (SDG 2) and encompasses most other SDGs. However, health systems that deliver quality nutrition services to its people remain a global challenge, especially in low- and middle- income countries (LMICs). Challenges exist in coverage and quality of services, compounded by ever-increasing healthcare costs [1]. Global experiences in the backdrop of sub-Saharan Africa, South East Asia and Latin America, however, illustrated significant contributions of community health workers (CHWs), the frontline primary health service providers, in LMIC settings [2]. Even with a large shortfall of human resources against the enormous population of beneficiaries, CHWs reached both mainstream and disadvantaged populations for service delivery [3,4,5,6,7]. This workforce, therefore, must be strengthened by promptly addressing identified challenges and obstacles for them, such as by providing necessary and service-appropriate equipment as well as regular and efficient supervision [3,8]. With widespread expansion of personal mobile phone usage and internet coverage, along with declines in associated costs, comprehensive and customised application of mobile technologies can provide significant assistance in health services delivery [9]. Termed as ‘mHealth’, this platform mostly makes use of hand-held mobile devices such as smartphones, tablets, personal digital assistants (PDAs) and portable computers, set with customised software applications [10,11]. Benefits and advantages of mobile devices and platforms include portability, fast and immediate information delivery, tracking capabilities, and easily adaptable features [12,13]. Thus, best matches considering service, device and software need to be explored for assisting in health services delivery. Most commonly documented mHealth-based interventions include text messages and phone reminders to improve uptake of health services including antenatal and postnatal care, and immunisation [14,15,16,17]. Mobile texts may help to curve healthy behaviour for the short term, but sub-optimal literacy rate, marked difference in equity of possessing mobile phones, and low utilisation of mobile phones for health services during pregnancy and childbirth remain major concerns, especially in Bangladesh [17]. These concerns reduce the reach of target groups for nutrition interventions which significantly comprise poor women. Moreover, interactivity between beneficiary and service provider generates greater interest; an aspect especially important in behaviour change communication (BCC) including counselling, where mHealth can be used as a job aid [10]. Moreover, Labrique, et al. [18] identified 12 mHealth functions that have been applied in health systems to include the following: (1) Client education and BCC, (2) Sensors and Point-of-Care Diagnostics, (3) Registries and Vital Events Tracking, (4) Data collection and reporting, (5) Electronic health records, (6) Electronic decision support, (7) Provider-to-provider communication, (8) Provider work planning and scheduling, (9) Provider training and education, (10) Human resource management, (11) Supply chain management, and (12) Financial transactions and incentives. The authors consider mHealth as a vehicle to strengthen health systems, calling for continuous evidence generation to understand impact of mHealth. However, Tonkin, et al. [19] mapped mHealth applications aiming nutrition behaviour change, concluding that all identified studies were conducted in developed countries, had lower sample sizes, showed scarcity of ‘social connectivity’ features, and lacked a comprehensive description of applications, while others found mixed results of applying mHealth in community settings [19,20]. 

Recent mHealth interventions in LMICs include a ‘mobile calculator’ of weight-for-height/length z score [21] and a tablet-based app addressing CMAM in 5 developing countries [22]. Creating such an enabling environment motivates staff, eases their tasks, and improves task implementation [23]. Yet, publications describing nutrition intervention delivery system along with its benefits and barriers are relatively few, which otherwise could provide important assistance to stakeholders to understand and develop similar platforms with greater feasibility. Researchers have proposed frameworks and guidelines for developing BCC interventions, such as COM-B (behaviour change wheel) and behaviour-centred design (BCD) approaches [24,25]. 

In a community-based cluster randomised controlled trial (cRCT) in rural Bangladesh, nutrition interventions targeting the first 1000 days of life (conception to 24 months of age) were provided to mother-child dyads as domiciliary services, which included time-specific, nutrition-focused BCC (counselling and practical demonstration) and lipid-based nutrient supplement (LNS) delivered by CHWs [26]. An Android platform-based app and web-linked system were developed to aid in implementing these interventions. This system applied 7 of the 12 mHealth functions described by Labrique et al. Our interventions improved exclusive breastfeeding till 6 months [27], and children’s dietary diversity at 6–24 months of age [28]. In this article, we: (i) describe the design and development of the tablet-web-based nutrition intervention delivery system; (ii) present the adequacy of intervention visit coverage from real-time data; and (iii) explore the facilitating and challenging issues of this system described by users (CHWs and their supervisors). 

## 2. Materials and Methods

### 2.1. Study Design and Participants

The study was set in *Hobigonj* district in *Sylhet* division, situated in the north-eastern region of Bangladesh. The area is a combination of level and elevated plains crisscrossed frequently by water bodies, making some of this place vulnerable during the rainy seasons. National surveys report this area with the presence of some food insecurity and a high prevalence of childhood stunting: 37% of households had mild to severe food insecurity while childhood stunting was 49% in 2011, before the study started [29]. Childhood stunting reduced to 43% by 2017 [17]. Hence, from two purposefully selected sub-districts (*Bahubol* and *Nobigonj*) of *Hobigonj* district, 12 unions (lowest administrative unit comprising of villages) were identified as eligible and divided into village-based clusters. Block randomisation was applied thereby with five or multiples of five clusters in each union. Two hundred and fifty pregnant women in each intervention arm (10 per cluster) and 500 pregnant women in the comparison arm (20 per cluster) were enrolled, totalling 1500 enrolled women, who delivered 1355 live births [26]. 

A detailed study protocol was published earlier [26]. Our study was built on the Lancet Maternal and Child Nutrition Series 2013, which identified 10 nutrition-specific interventions to reduce childhood stunting in resource-poor settings [30]. Their model-based assumptions called for primary research that tested the combined effects of multiple interventions provided as packages. Hence, we selected five preventive interventions from the list covering the first 1000 days of life, which include the following: (1) BCC on nutrition during pregnancy; (2) BCC on exclusive breastfeeding (EBF) for postnatal first six months; (3) BCC on complementary feeding throughout baby’s 6–24 months of age; (4) LNS for mothers during pregnancy (prenatal LNS, or PNS); and LNS for children during complementary feeding age of 6–24 months (complementary LNS, or CNS). However, we considered the three BCC interventions as a continuum under ‘BCC on maternal and child nutrition throughout pregnancy to postnatal 24 months’, resulting in three basic interventions (BCC, PNS, and CNS). Thereafter, we developed four combinations of ‘intervention bundles’ to test the effect of each bundle in addressing childhood stunting and other indicators of childhood nutrition. For testing our hypothesis, we designed a cRCT with mixed-method approaches. Briefly, this cRCT had five arms: four interventions and one comparison arms. In intervention arms, nutrition intervention bundles were delivered to mother-child dyads through CHWs who provided BCC and LNS in door-to-door visits, making use of the mHealth-based job aid’s various functions. Each visit took around an hour, but sometimes more if the mother was busy with her household chores. Other family members and interested female neighbours were always encouraged to listen to the BCC messages. However, BCC was never provided over the phone, and mothers never had to visit the study office or any facility to receive the interventions. If a mother was absent at home during CHW’s scheduled visit, CHWs contacted family members who were present at home and/or mothers over phone for their availability, and re-visited mothers at home at a later date. The BCC components were adapted from the ‘Key messages booklet on the community infant and young child feeding counselling package’ by WHO-UNICEF [31], with linguistic support from validated Bangla BCC modules [32,33] [Table 1]. CHWs also provided practical demonstrations of relevant messages, such as position and attachment for proper breastfeeding. Messages in the job aid were in the native language, Bangla. As for the LNS, selected micronutrients were incorporated in peanut paste and lipids, considering specific needs of the pregnant mother and young child. Therefore, the final intervention combinations (‘bundles’) were as follows: arm (1) BCC + PNS + CNS; arm (2) BCC + PNS; arm (3) BCC + CNS; and arm (4) BCC only. The comparison arm (arm 5) received only government-led routine health and nutrition services (Table 2). We had two separate field teams for implementing the study, each with two layers of supervisors: (A) the intervention implementation team consisting of CHWs delivering home-based BCC and LNS; (B) the evaluation team consisting of enumerators collecting data for evaluation. Such separation of teams ensured minimal bias in data collection for intervention evaluation. The tablet app system had functions for the CHWs who provided door-to-door intervention to the mother-child dyads, and for the enumerators who collected door-to-door evaluation data from mothers. CHWs’ work process is detailed in next sections.

### 2.2. Architecture of the Electronic Intervention System 

We illustrated the BCC messages through attractive texts and images, set into an application to be used by CHWs while counselling and demonstrating to mothers. Additional aims included record keeping of CHWs’ visits in real-time, and LNS distribution. We developed an architecture of ‘tablet app-based data collection, with an internally hosted application server’, which mitigated security concerns, and was highly suitable for setting our intervention modules and data collection questionnaire because of its advantage of customisation [34]. The programming language was Java. Data in mobile devices was stored in SQLite. After uploading, data was stored in SQL server 2008R2. 

Usage of the electronic system (Figure 1) was initiated after eligible pregnant participants were identified through paper-based, door-to-door surveys by enumerators. Primarily identified participants took a urine test confirming pregnancy, and eligible pregnant participants were enrolled on the study. Uploading this information to the server registered the pregnancy in the database. The app assigned a unique ID to each enrolled participant. CHWs could then download enrolled mothers’ list in their customised tablets. The algorithm built into the system used each ID’s first day of last menstrual period (LMP) date to calculate and auto-generate all intervention visit schedules during the pregnancy period till the associated delivery outcome is registered. Similarly, entry and downloading the date of birth (DoB) of the resultant baby generated unique IDs for each child and auto-generated postnatal intervention visit schedules. Similarly, enumerators’ tablets auto-generated evaluation schedules, linking with relevant digital questionnaires. Notably, CHWs’ tasks interlinked with enumerators only at two-time points: entry of LMP and DoB. 

Before initiating BCC sessions, the intervention team received intensive training that included comprehensive components on tablet and app usage and troubleshooting. 

#### 2.2.1. BCC Modules

The major topics of all pre- and post-natal BCC modules are listed in Table 1. The BCC component contained visit-specific modules with nutrition-specific and some nutrition-sensitive key messages. Each visit-specific module showed a hyperlinked index of topics. Hyperlinks directed to one or more ‘images’ [Figure 2a,b], each containing concise messages presented as short texts and pictures. Using the app as a job aid during home visits, CHWs went through each indexed topic, and freely navigated among the topics according to conversation flow. Each pregnant mother was entitled to 4 antenatal BCC visits, each live birth entitled the mother-child dyad for 13 postnatal BCC visits. Under each participant’s name in the app, full visit sets were presented as ‘button(s)’. Colour codes indicated visit status [‘Inactive’, ‘Scheduled unattended’, ‘Incomplete’, ‘Completed’; Figure 3].

#### 2.2.2. LNS Inventory

Two separate simple templates were developed in the app to track LNS distribution in relevant clusters. The template had spaces to input numbers of unopened supplement sachets mothers had, empty sachets consumed by the target subject, sachets they admitted to have destroyed or shared with others, and new sachets distributed by CHWs on that visit. 

### 2.3. Intervention Implementation and Monitoring with the App

Four clusters were fixed per CHW. Accordingly, each CHW received a customised tablet with prepaid data package, fixed catchment area, and participants assigned to her not overlapping with other CHWs. Both the tablet and the app were provided to the CHWs free of cost, as project assets. They could filter respondents according to place of residency (union and village) through a drop-down menu for planning their day-to-day tasks, considering urgency and convenience. We widened the initial narrow timeframe for intervention delivery to approach more mothers who otherwise could not be reached due to the prevalent cultural practices of pregnant women visiting their parents’ home during pregnancy and childbirth, and seasonal excursions. Every morning, they logged into the app and downloaded all data to update their visit task list and informed their schedule decision to the Field Supervisors before commencing daily visits. When CHWs logged into the application from mother’s home, date and time of each visit were recorded on tablets in real-time. CHWs could also use the app offline. Essentially, they uploaded all real-time and collected data every evening, synching with the central server online. Online synching also updated the app. 

The interventions (BCC and LNS, per design) were provided by the CHWs to the mothers during household visits (as domiciliary services). Mothers never had to visit the study office or any facility to receive the study intervention(s). CHWs visited mothers guided by the autoschedule function of the tablet-app system. Hence, irregularities were faced only when mothers were absent from home. In such cases, CHWs could input the reason(s) of the unsuccessful visit in the tablet. They also communicated with family members who were present at home and/or mothers over phone for their availability, and re-visited mothers at home at a later date. The visit was flagged in red until it was conducted properly. If a certain visit could not be conducted even after multiple attempts and the visit window expired, the visit button remained red. Interventions were never provided over phone. Mothers’ short-term absence from home was easy to address with such re-visits. For the few mothers who had migrated to places outside of the study area, we considered them drop-outs as per the protocol. Phone communication with the mothers was thus used only for discussing their availability at home, and never for conducting the sessions. This was because we considered face-to-face BCC more effective, especially for pregnant and nursing mothers in rural areas, and LNS had to be delivered in person. The only time we faced hurdles for mothers’ long-term absence was during delivery when mothers were either in the hospital or went to their parents’ home far from the study area (visit for ‘PNV within 48 of birth’). We have discussed this issue in later sections.

We developed a web-linked monitoring system or dashboard with several windows, which could be accessed by the field supervisors and the central team from desktops set with the system [Figure 4]. Specific stored information such as scheduled intervention visit dates, women’s LMP and children’s DoB, visit completion status—in short, information required for staff monitoring could be accessed through this desktop system. This information could be filtered union-, village- and cluster-wise. Supervisors could comprehend the overall and individual status of visits covered, time-outs, and reasons of missed visits, and make queries to the responsible CHW/FS accordingly, from their laptop.

### 2.4. Data Collection

The primary outcome variable for this article was coverage of CHWs’ visits throughout the intervention period, collected from the application-generated dataset, indicative of timeliness of CHWs’ scheduled visits (quantitative data). At the start of each visit, using the digital visit status form, CHWs entered information on the participant mother’s presence, consent, pregnancy status and/or baby’s condition during that visit; dates in cases of stillbirth, baby’s or mother’s death and reasons if mother was absent. Exclusion criteria for receiving antenatal visits included any pregnancy outcome (miscarriages/abortions, stillbirths, and live births), mothers’ death, and permanent emigration. The tablets auto-recorded the time CHWs entered their visit status (during home visits), and we could check the time stamp to ensure data quality. More importantly, two layers of supervisors monitored their tasks in person; thus, every CHW was monitored by a supervisor, at least, once a week. Weak CHWs received additional supportive supervision on identified issues. Hence, digital data collection along with direct human supervision ensured better quality of intervention and data while reducing biases and limitations. For this paper, we analysed data of the 1000 enrolled pregnant women and the resultant babies across the four intervention arms. 

Additionally, a qualitative researcher, independent of the study team, conducted qualitative interviews to gain insight into user experience of the application, i.e., CHWs’ and their supervisors’ perceptions and attitudes on benefits, challenges and possible solutions of delivering nutrition intervention using tablets and the software/app. The qualitative researcher conducted five in-depth interviews (IDIs) with CHWs, one focus group discussion (FGD) with Field Supervisors, and one IDI with the intervention team supervisor. From 23 CHWs, data saturation reached at five IDIs; hence, no further IDI was conducted. Field Supervisors were six in number and all joined the FGD. The only intervention team supervisor gave an IDI. To capture maximum in-depth information, CHW selection criteria included distance between the working area and CHWs’ residence, cluster coverage, recruitment period, CHWs’ age and marital status. Besides, qualitative data was collected near the end of the study period to ensure all benefits and challenges faced by the team is shared by the staff with limited risk of bias. Thus, data saturation was ensured and maximum data was captured on the topic. Our focus was on users’ attitudes and perception on the job-aid, and study mothers were not interviewed. 

### 2.5. Data Management and Analysis

Quantitative data was first cleaned to remove data entry errors and outliers. Visit coverage is presented as frequencies and percentages. Visits were considered timely (‘within schedule’) if it was provided within 7 days of the scheduled date, and ‘beyond schedule’ afterwards. For example, timely visits were calculated as child’s age in months ±7 days, all others were considered beyond schedule. Child’s age was calculated as the difference between DoB and date of the visit. Considering all intervention arms received BCC, we combined all intervention arms, and a descriptive analysis was conducted. We analysed visit coverage and reasons for drop-out. We conducted chi-square tests to analyse categorical data and considered differences in outcome as statistically significant at *p*-value < 0.05. Stata SE 15 was used for data cleaning and analysis. 

Qualitative data was analysed employing hybrid thematic analysis. Initially, an a priori codelist was developed based on interview guidelines. After conducting qualitative interviews, audio recordings were transcribed verbatim in Bangla, enriching with field notes. The a priori codelist was used as primary codes and assigned to interview transcripts, adding new codes along the process. Next, closely matching codes were narrowed down to sub-themes, and themes emerged from sub-themes. Finally, analysis and interpretation of themes and sub-themes were carried out to prepare the final analysis. Relevant and illustrative quotes were added to enhance interpretation. All qualitative data was managed and analysed manually, and coded quotes and quotation summaries were managed using an Excel workbook (MS Office standard 2019).

### 2.6. Ethics Approval

We obtained written informed consent (which included full disclosure on the study) from each participant twice: during initial enrolment at pregnancy, and during enrolment of resultant live births. Privacy, anonymity and confidentiality of the information provided by respondents were strictly maintained at all phases of the trial. All information were stored in an encrypted database with participants’ study ID instead of personal identifiers, and none but the associated investigators and data management team had access to the collected data. We conducted the study in accordance with the Declaration of Helsinki. The ethical review committee of icddr,b approved the trial protocol. We registered the study at ClinicalTrials.gov (ID: NCT02768181) (accessed on 1 July 2024).

## 3. Results

### 3.1. Intervention Visit Coverage: Quantitative Assessments 

For this article, we analysed visit data of all 1000 pregnant women from the four intervention arms (250 per arm). All pregnant women were eligible to receive four antenatal visits (ANV). ANV1 was provided within 15 days of recruitment to all enrolled women (within gestational age/GA 149 days), except one who was absent. Subsequent ANV coverages were 96–99% [Table 3]. Identified drop-out reasons during antenatal the phase included miscarriages/abortions, stillbirths, post-recruitment non-compliance (non-consent), and permanent relocation beyond the study area [Appendix A]. Live births reported during ANV3 and ANV4 entitled and forwarded these mothers for postnatal visits (PNVs). Immediately after birth, visit coverages were: 52% (PNV within 48 h of birth), and 91% (PNV 7–14 days). Subsequent monthly visit coverage was very high at 96–99% throughout PNV 1–6 m. Afterwards, visits were provided every 3 months, and coverage remained high at 99% at PNV 9 m, 12 m, 15 m, 18 m and 21 m. Dropouts during the postnatal phase included stillbirths, child deaths, post-recruitment non-compliance, and permanent relocation. Mothers were marked as ‘permanently relocated’ when they relocated outside the study area that was too far for the CHWs to visit. We also explored arm-wise visit coverage, which showed no statistically significant differences across the arms [Appendix A]. 

We further categorised timelines of visits (Figure 5). Timely visit coverage (visit window + 7 days) was 96–100% throughout ANVs but dropped to 52% (46–55%) at PNV within 48 h of birth, which was explained by mothers staying at the hospital or their parents’ home for childbirth. When they returned home, visit coverage increased to ~88% during PNV at 7–14 days of birth, reduced again to 83% (79–86%) at PNV 1 m, but steadily kept going up. From 6 m to 21 m, visit coverage was mostly >90%. Inter-arm variation of visit coverage was low and statistically not significant. 

We analysed continuum of care, i.e., mother-child dyads receiving subsequent visits, although we did not adjust for eligibility. We found that 79% of all mother-child dyads received 14 more visits out of total of 17. This coverage would increase further if eligibility is considered and drop-outs are adjusted. 

Reasons for unsuccessful visits varied slightly across the schedules [Figure 6 and Appendix A. Apart from absence, some mothers were non-compliant, and some of them debarred the CHWs to visit them, although the number was very low.

### 3.2. Benefits, Challenges and Solutions from Healthcare Providers’ Perspective: Qualitative Findings

CHWs and their supervisors discussed benefits and challenges of using the tablets and software for delivering interventions, along with some troubleshooting to overcome the challenges. Benefits and challenges both could be broadly categorised into three types: technical (relating to the device and application), functional (relating to person-specific issues) and others. 

**Technical benefits:** Several CHWs mentioned that the software-based modules assisted them in remembering the counselling topics easily, which were taught during training, especially as visit-specific messages were set separately. Pictorial demonstration helped both CHWs and mothers to understand the messages and techniques. For example, CHWs could better demonstrate breastfeeding techniques, examples of unacceptable breastmilk substitutes and harmful junk foods, newborn babies’ small but growing stomach capacity, and other health messages were aided by pictures, while mothers appeared encouraged and to better understand from the pictures. A CHW stated: 


*“Mothers understand (messages) faster from pictures than verbal explanation”*

*[IDI, CHW]*


Using the native language was another benefit as mothers could read the messages themselves in pictures. CHWs also noted that tab-based information entry (i.e., for enrolment, visit-specific consent and LNS inventory) made their tasks easier and faster compared to paper-based approach, with less equipment to carry during visits. 

Automated (instead of manual) scheduling was another major benefit mentioned by multiple CHWS. Readily available full mother-child list, with their addresses along with set visit plans, enabled CHWs to plan daily schedules within a short time, make flexible visit plans, follow-up for birth delivery, identify and conduct emergent and critical visits (e.g., postnatal visit within 24-h of delivery), find listed but unknown households (for newly appointed CHWs), and distribute supplement packages conveniently; in general, creating a less stressful workload. Colour codes assisted them to easily understand visit status and plan accordingly. A CHW summarised: 


*“Workload seems much less because of using the tablet”*

*[IDI, CHW]*


Nevertheless, supervisors made daily home visits to observe CHWs’ intervention quality (message accuracy and counselling skills), tracked CHWs’ daily schedule, and conducted urgent BCC sessions if the corresponding CHW was absent. 


*“I have to ensure counselling quality: I check if important messages are overlooked, or if messages are clearly relayed to mothers … I check if the Field Supervisors can properly supervise and share feedback to their assigned CHWs … I supervise both these cadres”*

*[IDI, Intervention team supervisor]*


**Functional benefits:** Several CHWs identified tablet functions to be similar to that of smartphones, hence learning to operate the software was easy for them following the training phase and initial days into starting home visits. 

**Technical challenges:** Several CHWs reported technical challenges that arose from erroneous entry of the two dates that autogenerated subsequent fixed visit dates: first date of women’s LMP, and children’s date of birth (DoB). Enumerators and CHWs applied multiple techniques to help women recall correct LMP dates during enrolment and ANV1. However, some women did not keep track of their LMPs. Hence, some LMP dates were found conflicting when CHWs’ and enumerators’ probing was compared, while some mothers were self-motivated to inform corrected dates (e.g., from ultrasonogram reports) during CHW’s later visits. Incorrect dates invalidated benefits of autogenerated schedules. 

Considering DoB, some births went unnotified despite adopting several notification techniques. Late notifications came from mothers delivering outside the study area or in hospitals, including for caesarean section deliveries. A CHW explained: 


*“A mother maybe in hospital probably having a caesarean section, or the baby was in hospital with a problem, or the mother had a problem. She forgot (to notify birth), maybe she called us after five days and said that her baby was delivered but they could not inform us for this and that reason”.*

*[IDI, CHW]*


On an urgent basis, we crosschecked and updated the corrected dates while the team programmer updated LMP or DoB dates for rescheduling subsequent visits. However, CHWs had to conduct a few visits without capturing track in the system if visit dates would pass before system correction. 

Charging tablets was another major challenge. Battery capacity deteriorated over the course of time, while daily updating of the system consumed a higher charge. Eventually, devices slowed down from regular, continuous usage, so that even fully charging overnight could not ensure day-long battery coverage. A CHW pointed out: 


*“You can work with a charged tablet. (But my tablet) charge came down to 60–50% just after one visit, and turned off when I signed into another visit”.*

*[IDI, CHW]*


Although discouraged, charging at respondents’ homes was a temporary solution, which was, again, impossible during regular power cuts. Power cuts aggravate in Bangladesh during the rainy season. We replaced non-functional chargers, but it was unfeasible to change partial or whole sets of devices frequently. Working with tablets became practically impossible without electricity: 


*“Some places have no electricity. Even if I carry the charger, I cannot use it if the family uses (low energy storage) solar power … so we always keep hard copies (of BCC messages) and use it as alternatives”.*

*[IDI, CHW]*


Instability of internet network especially during rainy days, heavy fog, windy days and bad weather, and in hilly areas, further complicated by undercharged tablets, severely hampered downloading and uploading the system, and using BCC online:


*“Many times, it does not upload or download, (or) takes a long time … say, it keeps uploading for an hour … it takes forever to upload”*

*[IDI, CHW]*


Long-term usage often caused tablets to ‘freeze’ (not responding), generally solved by restarting multiple times, but a cumbersome task during working hours. Sometimes dates within the tablet changed, or tablets did not accept passwords to log into the app, requiring urgent troubleshooting from the programmer. 

Hence, all staff were trained on simple troubleshooting, enabling them to fix common problems themselves. For persisting problems, they informed supervisors immediately. CHWs acknowledged that their supervisors were always available to troubleshoot on-site: 


*“(We were given) simple instructions: like, take the tab to an open space, take some time and be patient. If it still doesn’t work, shut it down, wait for some time and turn it on again. Wait half an hour. If not fixed by this, wait another half an hour. If much time is gone but the problem persists, we take the tab to office”.*

*[IDI, CHW]*


Time-tracking of topic-specific counselling emerged as impractical during free-flowing counselling sessions, especially since mothers frequently remained engaged in household chores. Quality counselling was more challenging during harvesting seasons. 

**Functional challenges:** Only one staff reported erratic finger control over touchscreens, which she overcame with continuous usage. 

**Other challenges:** Tablet safety was a major concern both during working hours and at home, especially for those with little children who are attracted to screens. One CHW accidentally broke her tab. 


*“I am continuously worried if something happens to the tab, since it is an official asset”.*

*[IDI, CHW]*



*“I’m always scared if the tab is lost … at work and at home … we work alone so it can be hijacked”*

*[IDI, CHW]*


One CHW acknowledged that CHWs may use official tablets for personal causes, which can contribute to quality and longevity deterioration. 

Throughout the three years of study period, 8 of 24 tablets had to be urgently replaced; reasons include being broken screen, tablets falling into water, tablets getting stolen, and severely malfunctioning. 

## 4. Discussion 

To our knowledge, this study was among the first in Bangladesh employing a paper-less, detailed and customised, fully documented mobile technology-based system in a community-based intervention study in rural Bangladesh for delivering nutrition counselling, tracking LNS, guiding CHWs with auto-scheduling function for home visits, and providing system-based monitoring aid to supervisors. This article illustrates the design and development steps of the intervention and monitoring system, with advantages and challenges of this platform described by the users (CHWs and their supervisors), adding to the meagre literature available on this evolving platform. We employed the ‘tablet app-based data collection, with an internally hosted application server’ architecture. Although costlier, it provided the advantages of system customisation and ensured data security. Keeping both intervention and assessment modules in the same tablets would have created substantially large files [35], so tablets were designed staff-specific. Apart from reducing file sizes, this eliminated the chances of the field staff confusing or mistakenly entering into another staff’s assigned respondents and tasks. Advantages of using tablets over commonly used mobiles includes additional functionality, ability to handle more complex data, and bigger screens [36]. 

We documented that with appropriate training and supervision, locally recruited CHWs could properly use a tablet-based app for on-the-spot entry of their visit status, capturing reasons and associated information for unattained visits, thus registering their daily tasks of door-to-door visits which could be retrieved for analysis and fine-tuning interventions. CHWs attained high coverage of both scheduled visits and continuum of visits. Previous studies support accuracy, easier workload management and staff preference of device-based data collection [36], employed in large-scale DHS data collection [37]. 

While our CHWs provided most counselling sessions within time, several socio-demographic factors impeded full intervention coverage. A major practical challenge was the widespread cultural practice of pregnant and nursing mothers moving from their marital home to their parents’ home for better home-based care during pregnancy, childbirth and nursing, for different durations. This, along with delivery in the hospital, caused lower number of contacts especially during the critical peri-natal period. This culture is commonly reported throughout South Asia [38]. Hence, contacting ‘relocated’ and/or traveling pregnant and nursing mothers must be addressed while designing similar but large scale, regular programmes, such as through keeping track and linking their careseeking history. On technical grounds, dependency on internet for delivery information uploading-downloading-scheduling created tension during this critical period because of frequent internet disruption. In case of most other visits, however, CHWs could utilise the advantage of using the system offline and uploading/synching data at a later time when internet was smooth. This approach is feasible in most rural areas and can be replicated.

Mobile-based intervention mostly employs one- or two-way text messages as reminders [14,15], while direct human interaction during intervention delivery, for example BCC through ‘peer educators’ has a better effect on beneficiaries in contrast to indirect, mechanistic approaches [39]. Therefore, for strengthening mHealth as a strong tool of intervention delivery, our modules were adapted fully in Bangla and enriched with images, which was highly appreciated by CHWs and mothers. The BCC index was designed for easy navigation that accords with the need and flow of the counselling session without breaking the flow of interaction between CHW and mother, and advantage of the bigger tablet screen was also availed. Earlier, Murray (2005) showed that the ‘interactive health communication applications (IHCA)’ approach combining computer-based health information packages with online supports from peers shows positive behaviour changes and confidence among patients with chronic diseases [40]. In educational sector, computer-assisted instruction (CAI) is progressively being used as an interactive, personalised approach in school-based teaching-learning [41], which may have similar impact on health counselling delivery. 

Applying Labrique et al.’s mHealth framework [18], seven of the 12 mHealth functions were incorporated in our system: (a) Client education and BCC (e.g., BCC modules), (b) Registries and Vital Events Tracking (e.g., LMP and birth registration at home), (c) Data collection and reporting (e.g., CHWs’ visit status), (d) Electronic decision support (e.g., visit-specific BCC modules, auto-scheduling of CHWs’ ANV and PNV visits), (e) Provider work planning and scheduling (e.g., auto-scheduling of CHWs’ ANV and PNV visits), (f) Provider training and education (e.g., BCC cue to CHWs during home-based sessions), and g) Supply chain management (e.g., LNS inventory form). ‘Data collection and reporting’ function was also applied for data collection for intervention evaluation by the enumerators of the evaluation team, but this article is beyond the scope of their activities. 

We addressed cultural aspects of the interventions and the job aid, both in terms of facilitators and barriers. Globally, home-based BCC is a widely accepted nutrition intervention while LNS has been tested in multiple studies across the LMICs without objection by beneficiaries, and all these interventions were listed as effective in the Lancet Maternal and Child Nutrition Series 2013 [30]. Usage of devices similar to tablets (e.g., smartphones) has become common across the world. During the initial stage of developing the job aid, we explored if the images (adopted from the national CHW manual) were acceptable to the CHWs and mothers, both of whom had no objection. Throughout the study period, CHWs reported negligible disapproval from mothers about the images (source: informal field notes). While studies have explored various cultural barriers in LMICs against IYCF practices, intervention studies proved that home-based BCC intervention significantly improved IYCF practices, including some recent studies employing mHealth components [42,43,44,45,46]. 

There are 2.26 community and domiciliary health workers per 10,000 population in Bangladesh, whose tremendous workload needs assistance. Public health researchers expressed that addressing occupational stress of CHWs should be an integral part of the intervention package [47]. Auto-generated daily task schedules and advanced availability of schedules reduced our CHWs’ workload and associated professional stress, increasing their efficiency in time management. Portability was also mentioned as an advantage of tablets as a job aid. In India, usage and adherence of mHealth-based job aid were higher among ASHA workers than among mainstream PHC healthcare staff [38], calling for rigorous exploration of such differences, before wider scale-up of mHealth as job aid. 

Nevertheless, ‘regular and continuous supervision and monitoring systems’, identified as part of performance management of the CHWs [3], were not undermined in our study. Our supervisors ensured quality of counselling, both-way interactions between mothers and CHWs, and customising quality of counselling as needed. We previously showed that in our study area, BCC coverage was very low through public healthcare system, and that our BCC intervention significantly improved multiple indicators of breastfeeding and complementary feeding [27,28]. Thus, mHealth did not replace nutrition service providers, but was applied as a support to them for smooth running of their tasks. 

We highlight system-related challenges also. Erroneous LMP and DoB entries caused major mishaps in auto-generated visit scheduling, which required prompt corrective actions. This issue should be addressed efficiently in large scale programmes. While implementing similar apps, rigorous pilot testing is crucial along with regular cross-checking conducted by project staff [36]. Nevertheless, updates would be needed periodically. Major technical problems while mainstreaming mHealth, especially in resource-poor settings, such as frequent power-cuts, deteriorating device and battery quality, and disrupted internet connection causes interruption in smoothness of workflow. Data security of a stolen tab remains a major concern. King, Buolamwini, Cromwell, Panfel, Teferi, Zerihun, Melak, Watson, Tadesse, Vienneau, Ngondi, Utzinger, Odermatt and Emerson [36] pointed out similar concerns and noted that designing data collection should be ‘within the limits of the local infrastructure’ while security is of utmost importance. Bangladesh recently [48] adopted the DHIS-2 platform for collecting and utilising health data by government CHWs employed under the health systems. Paper-based registers are being converted to the electronic system at multiple layers. Even grassroot health cadres receive official devices and internet connectivity for various official activities. Job aid employing mHealth designed similar to the one we developed can be integrated (adopted or replicated) into this developing DHIS-2 platform, and therefore has the potential to support CHWs and their supervisors providing maternal and child health services [49]. However, paper-based copies should be kept at hand at all times so that emergency situations such as IT outage [48] do not hamper service delivery. 

Finally, concerns of occupational hazards from electronic devices are emerging: retinal damage from exposure to blue light emitting from devices [50,51] and the ever-increasing burden of e-waste have been documented [52,53]. Replacing broken devices is largely a funding concern, while smooth running of mHealth with frequent device replacement would inevitably add to environment damage, rendering this intervention a double-edged sword. Rigorous scientific studies should be carried out to understand the impact of mHealth on health and environment. Interventions should be designed and tested to regularly inform health service providers on device care to lengthen device durability, health hazards of continuous usage of electronic devices, and means to reduce negative exposure. Effective e-waste management should be scientifically explored and nationally scaled up. If ethically maintained, ensuring effective e-waste management would create new job opportunities. 

**Limitations:** Considering our study design, we could not compare effect of tablet vs. paper-based job aid among our CHWs. However, usage differences of health information set in electronic vs. printed material, mobile vs. stationary gadgets, and customised vs. generic interface appear to have low impact in smaller sample studies involving health practitioners involved in patient care [54]. 

As we aimed to examine only feasibility of the job aid in-depth, evaluating it applying any framework was beyond our scope. Nationally scaling up this or similar job aid and app, and future studies designing app addressing other health interventions, both can benefit by evaluating their intervention by applying comprehensive frameworks such as the COM-B and BCD approaches [24,25].

Another limitation of our analysis is the absence of cost-benefit analysis of the tablet-app system, as this was beyond the scope of our study. Future mHealth studies should explore on economic aspects of applying mHealth in nutrition interventions. However, Bangladesh and many other LMICs have been piloting and scaling up digitalisation of the health systems informatics for some years now [18]. Budget and funding are allocated by national and international stakeholders. Hence, mHealth approaches in LMICs should be rigorously evaluated including for cost-benefit analysis, and transparency should be maintained. 

## 5. Conclusions

Addressing the gap in published literature identified previously by authors, we narrated the architecture of a nutrition-focused job aid employing mHealth. Our analysis of feasibility of the system shows that with suitable training, CHWs can utilise this job aid proficiently for delivering nutrition intervention while keeping electronic track of their tasks, which their supervisors can retrieve for supervision and monitoring. Continuum of care was warranted over successive visits. Therefore, we conclude that this application and job aid can be incorporated into the existing health systems platform for improving coverage and quality of preventive maternal and child nutrition services. Continuous supportive supervision by skilled supervisors should accompany to maintain CHWs’ BCC quality. Finally, rigorous studies should be conducted to assess undesirable health and environmental effects of mHealth before and after mainstreaming, effective interventions addressing device-induced health hazards should be designed and scaled up, and effective e-waste management must be ensured. 

## Figures and Tables

**Figure 1 nutrients-16-03429-f001:**
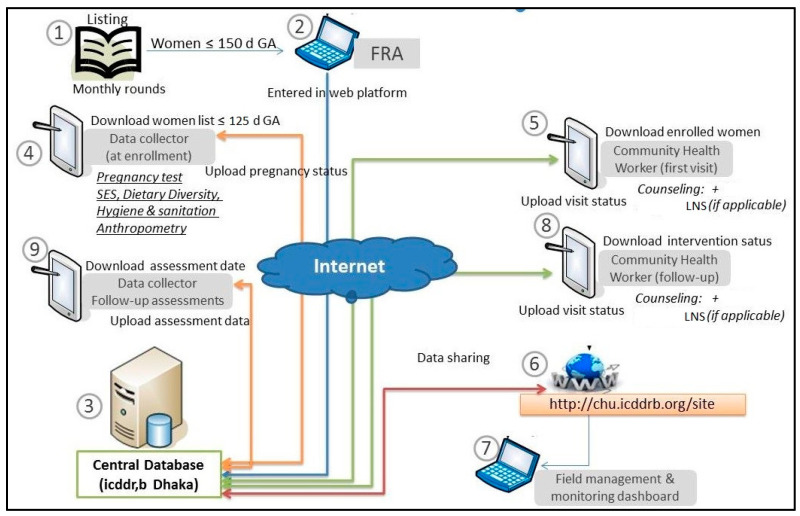
Data flow architecture.

**Figure 2 nutrients-16-03429-f002:**
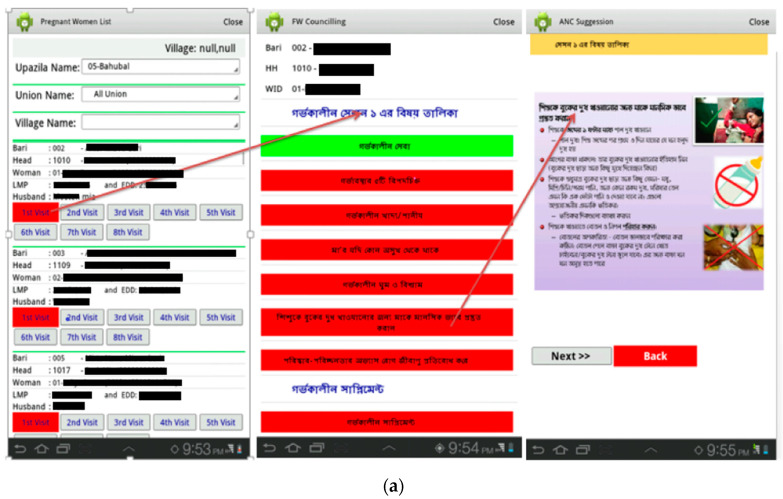
(**a**) Example of behaviour change communication (BCC) template (Bangla). (**b**) Example of BCC template (English translated). The left-hand image shows a partial list of all enrolled pregnant women under the sub-district ‘*Bahubal*’. Under each woman’s name and short identifications, all applicable intervention visits show in 2 rows of ‘buttons’. Clicking on a button would open another window showing major topics, also set as buttons, covered under that specific visit [e.g., middle image]. Clicking on a ‘topic button’ in this window would open another window showing short key messages [e.g., right-hand image].

**Figure 3 nutrients-16-03429-f003:**
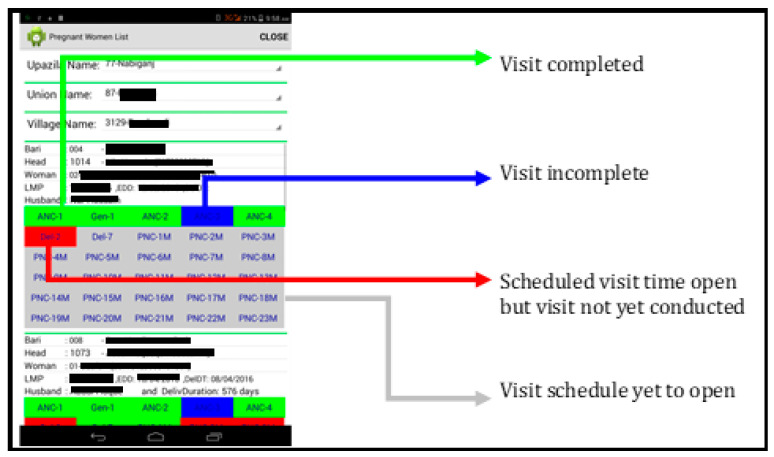
Colour codes used in the system.

**Figure 4 nutrients-16-03429-f004:**
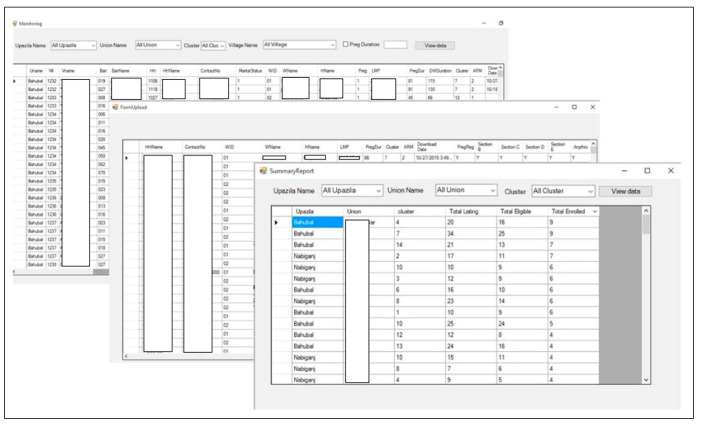
Monitoring windows.

**Figure 5 nutrients-16-03429-f005:**
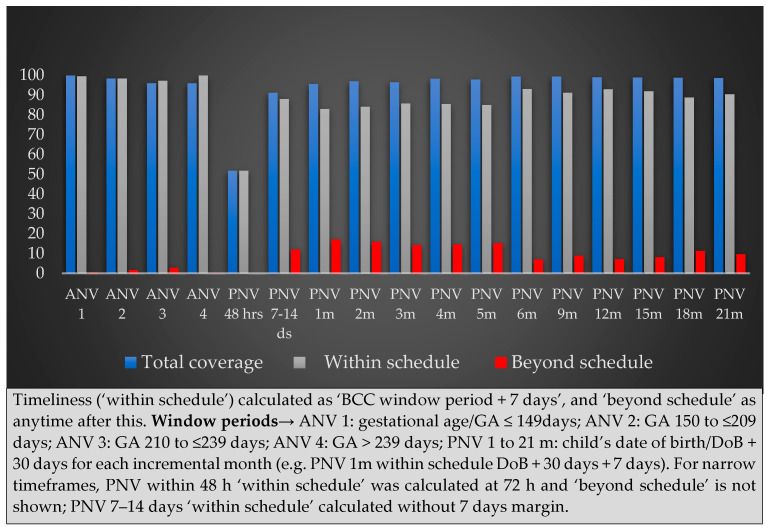
Timeliness of ANV 1 to PNV 21 m.

**Figure 6 nutrients-16-03429-f006:**
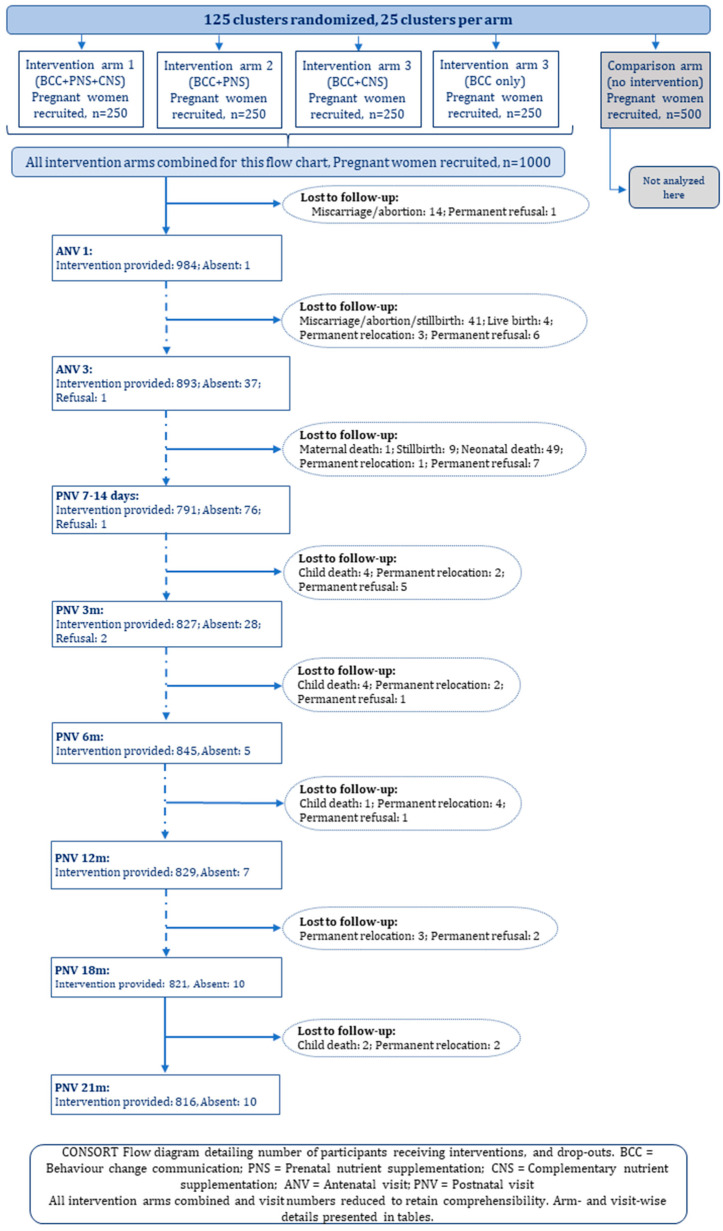
CONSORT Flow diagram.

**Table 1 nutrients-16-03429-t001:** Major topics of behaviour change communication (BCC).

Antenatal Visits (ANV) 1–4	PNV 6–11: 6–21 Months
Diet during pregnancy: Do’s and Don’ts (food groups, homemade good food, foods/drinks/tobacco to avoid, sample 1-day menu)	Homemade nutritious food + continued BF
Antenatal care (ANC), and birth preparedness	4 important food groups: animal protein, plant protein, complex carbs, fruits & veg, iron-rich foods, esp. animal sources
Pregnancy-related 5 danger signs and immediate care	Quantity: increases with age
Sleep and rest during pregnancy	6–8 m: ½ of 250 mL bowl; twice/day
Sensitising the mother for breastfeeding (early initiation of BF, avoiding pre-lacteal, exclusive breastfeeding/EBF for 6 m)	9–11 m: ½ of 250 mL bowl; thrice/day
Water, sanitation and hygiene (WASH)	12–23 m: full of 250 mL bowl; thrice/day
	Additional snacks: 1–2 times/day
**Additional counselling at ANV 2**	Responsive feeding
Anaemia: tackling through diet and hygiene	Practical demonstration of CF as needed
Extra care for teenage pregnancy	WASH during BF and CF
Preparing the mother mentally for neonatal care (drying and wrapping)	Feeding anorexic & sick infants: frequent, small feeds (BF, CF), increased food variety, avoiding junk food; caregiver’s patience
**Additional counselling at ANV 3–4**	Vaccination, care seeking during illnesses,
Birth preparedness at home	Diet for lactating mothers, sample 1 day menu
Position and attachment for breastfeeding/BF (practical demonstration)	Family planning, WASH, sleep & rest
Postnatal care (PNC)	
**Postnatal visits (PNV) 1–5: birth to 5 completed months**	BF essentials continued (position & attachment, expression etc, as needed)
Continued: EBF for 6 m, position & attachment (hands-on demo)	
Rules and signs of enough BF	
Expression of breastmilk; care for breast problems	
Caring the low birth weight, sick infants	
PNC, vaccination, home care for mother & child	
Danger signs of neonate, infant and nursing mother	
Diet of lactating mother, sample 1-day menu	
Family planning, WASH, sleep & rest	

**Table 2 nutrients-16-03429-t002:** Intervention distribution across the arms.

Intervention(s)	Intervention Arms				Comparison
	Arm 1	Arm 2	Arm 3	Arm 4	Arm 5
BCC					-
PNS			-	-	-
CNS		-		-	-

The colours represent presence of that particular intervention in the associated arm.

**Table 3 nutrients-16-03429-t003:** Coverage of ANV 1 to PNV 21 m (four intervention arms combined).

Intervention Visits	Eligible	Conducted
	*n* (%)	*n* (%)
**Antenatal visits (ANVs) [Enrolled women: 1000 in four arms combined]**		
ANV 1	985 (98.5)	984 (99.9)
ANV 2	950 (96.4)	934 (98.3)
ANV 3	931 (98.0)	893 (95.9)
ANV 4	868 (93.2)	832 (95.9)
**Postnatal visits (PNVs) [Resultant live births: 1355 in four arms combined]**		
PNV within 48 h of birth	874 (94.5)	453 (51.8)
PNV within 7–14 days of birth	868 (99.3)	791 (91.1)
PNV 1 month	864 (99.5)	826 (95.6)
PNV 2 month	858 (99.3)	832 (97.0)
PNV 3 month	857 (99.9)	827 (96.5)
PNV 4 month	852 (99.4)	837 (98.2)
PNV 5 month	852 (100.0)	833 (97.8)
PNV 6 month	850 (99.8)	845 (99.4)
PNV 9 month	844 (99.3)	838 (99.3)
PNV 12 month	837 (99.2)	829 (99.0)
PNV 15 month	834 (99.6)	825 (98.9)
PNV 18 month	832 (99.8)	821 (98.7)
PNV 21 month	828 (99.5)	816 (98.6)

## Data Availability

Data described in this manuscript are available on request from the corresponding author due to privacy and ethical reasons.

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
