# Peer review of "Feasibility of Employing mHealth in Delivering Preventive Nutrition Interventions Targeting the First 1000 Days of Life: Experiences from a Community-Based Cluster Randomised Trial in Rural Bangladesh"

_nutrients, 2024, doi:10.3390/nu16203429_

Round 1

Reviewer 1 Report

Comments and Suggestions for Authors

Dear Authors,

The manuscript submitted for review is interesting and presents the use of modern technologies to monitor the health of a pregnant woman and her child after giving birth and as a nutrition intervention. I wonder if the manuscript title, Feasibility of employing mHealth in delivering preventive nutrition interventions targeting the first 1000 days of life: Experiences from a community-based cluster randomized trial in rural Bangladesh, is appropriate.

The manuscript has little information about how the nutritional intervention was performed. It is not clear to me. More details are needed on this subject.

The number of people participating in the study is sufficient, and the study was planned correctly.

However, the text presented in lines 96-115 is not well prepared, and it is not fully understood what the experiment was about.

 Lines 145-146 – It would be necessary to explain what is presented in Figure 2 because the information is probably in Bengali.

Please explain what happened if irregularities were found, e.g., a woman skipped a check-up visit. Was there any telephone or other intervention? This is important because otherwise, it would only be a system for monitoring and collecting data. In my opinion, the description of this part of the methodology needs to be improved.

Readers should know more clearly what was done in the case of irregularities.

 It seems that more should be written about the Limitations of these results. Indeed, the limitation is the participants' possession of portable devices, such as smartphones, tablets, PDAs, and portable computers. Where the application was downloaded from and whether it was free is unknown. The authors also write about problems with charging tablets and issues with the internet in Bangladesh. Since we are talking about the rural environment, the Internet is a crucial topic because the problem is usually associated with a lower IT and telecommunications network density in the rural environment.

Conclusion – For such a large study, the conclusions are very enigmatic. They should be improved.

 Technical notes:

Line 2 – the word ‘Titl’e is not needed.

Figure 2 and Figure 4 - They are of poor  quality

Table 1 - The table should be formatted so that it is not stretched across three pages, which is unreadable for the reader.

References are not prepared by the Nutrients Journal requirements.

 I am pleased to recommend this manuscript for publication but after major revision.

Reviewer

Reviewer 2 Report

Comments and Suggestions for Authors

The title clearly conveys the research's focus, methodology, and context. It is informative and aligns well with the study’s content.

The abstract could benefit from more explicit emphasis on the study's broader implications for policy and practice, as this would increase the paper’s impact on future research and public health interventions.

The introduction could elaborate more on the theoretical framework guiding the study, such as providing a deeper rationale for the use of mHealth and its potential beyond convenience (e.g., the psychology behind behavior change communication).

Methods:

The description of the control and intervention arms, although detailed, can be streamlined to ensure clarity for readers less familiar with cluster randomized controlled trials.

More information on how qualitative data saturation was determined would improve the transparency and rigor of the qualitative analysis.

There is limited discussion on potential biases or limitations regarding the CHWs’ role in data collection, which could affect data reliability.

Results: The presentation of statistical data could be more concise, perhaps utilizing visual aids such as tables or figures to enhance readability.

Discussion: 

The discussion of limitations is somewhat limited. The potential bias introduced by CHWs' self-reported data could be addressed more explicitly.

The broader applicability of the study’s findings to other LMIC settings should be explored further, including possible cultural, technological, or economic barriers that may arise when scaling up such interventions.

There could be more critical analysis of the long-term sustainability of mHealth in resource-poor settings, particularly in relation to the costs and environmental impact highlighted in the study.

The conclusion could be expanded to suggest more specific recommendations for future research or practical implementation, such as potential improvements to mHealth platforms to address the challenges identified in the study.

Round 2

Reviewer 1 Report

Comments and Suggestions for Authors

Dear Authors,

The authors have significantly improved the manuscript based on the reviewers' comments. It was corrected and supplemented all sections of the manuscript.

Titles of Tables 1,2,3 should be above the tables.

I am currently recommending the manuscript for publication in the Nutrients journal.

Reviewer

Reviewer 2 Report

Comments and Suggestions for Authors

I am satisfied with the changes provided by the authors